# EFFICIENT SEQUENCE LABELING WITH ACTOR-CRITIC TRAINING

## ABSTRACT

Neural approaches to sequence labeling often use a Conditional Random Field (CRF) to model their output dependencies, while Recurrent Neural Networks (RNN) are used for the same purpose in other tasks. We set out to establish RNNs as an attractive alternative to CRFs for sequence labeling. To do so, we address one of the RNN's most prominent shortcomings, the fact that it is not exposed to its own errors with the maximum-likelihood training. We frame the prediction of the output sequence as a sequential decision-making process, where we train the network with an adjusted actor-critic algorithm (AC-RNN). We comprehensively compare this strategy with maximum-likelihood training for both RNNs and CRFs on three structured-output tasks. The proposed AC-RNN efficiently matches the performance of the CRF on NER and CCG tagging, and outperforms it on Machine Transliteration. We also show that our training strategy is significantly better than other techniques for addressing RNN's exposure bias, such as Scheduled Sampling, and Self-Critical policy training.

## 1 INTRODUCTION

Sequence labeling is a canonical structured output problem, with many techniques available to track its chain-structured output dependencies. Conditional Random Fields (CRF) built over a neural feature layer have recently emerged as a best practice for sequence labeling tasks in NLP (Huang et al., 2015). Alternatively, one can track the same output dependencies using a Recurrent Neural Network (RNN) over the output sequence, similar to the decoder component in neural machine translation (NMT). Using a decoder RNN instead of a CRF has several potential advantages, such as simplified implementation, tracking longer dependencies, and allowing for larger output vocabularies. However, shifting from the CRF's sequence-level training objective to the decoder RNN's sequence of token-level objectives may lead to suboptimal performance due to exposure bias.

Exposure bias stems from the token-level maximum likelihood objective typically used in RNN training, which does not expose the model to its own errors (Bengio et al., 2015). RNNs are typically conditioned on gold-standard contexts during training (a procedure known as Teacher Forcing (Goodfellow et al., 2016)), while at test time, the model is conditioned on its own predictions, creating a train-test mismatch. As sequence-level models, CRFs are immune to exposure bias.

We set out to establish the decoder RNN as an attractive alternative to the CRF for sequence labeling. We do so by adopting a simple and effective RNN training strategy to counter the exposure-bias problem, and by providing an experimental comparison to demonstrate that our modified RNN can match the CRF in accuracy, and surpass it in flexibility. Our chosen training method is Actor-Critic reinforcement learning (Konda & Tsitsiklis, 2003), which we adapt to the sequence-labeling scenario by providing immediate rewards after each output tag, and by letting those outputs adjust the critic scores. Our complete system, dubbed as the AC-RNN, maintains the same neural feature layer that has proven so successful for the CRF, changing only the output layer. We conduct a comprehensive analysis comparing the AC-RNN to the CRF under controlled conditions using a shared implementation. We demonstrate that on Named Entity Recognition (NER) and Combinatory Categorical Grammar (CCG) supertagging, the AC-RNN can match the performance of the CRF, while training more efficiently.

To demonstrate its flexibility, we also test our method on machine transliteration, a monotonic transduction problem that straddles the boundary between sequence labeling and full sequence-to-

sequence modeling. Finally, we compare our method with previous techniques proposed to address exposure bias. On NER tagging, we empirically demonstrate that AC-RNN is significantly better than the RNN trained with scheduled sampling (Bengio et al., 2015). We also demonstrate that our approach is more suitable for sequence labeling tasks than other similar policy-gradient methods such as self-critical training of Rennie et al. (2017).

## 2 PRIOR WORK

In sequence labeling, several neural methods have recently been shown to outperform earlier systems that use hand-engineered features. For the tasks of POS tagging, chunking and NER, Huang et al. (2015) apply a CRF output layer on top of a bi-directional RNN over the source. For the NER task, Lample et al. (2016) extend the RNN layer with character-level RNNs that capture information about word prefixes and suffixes. Ma & Hovy (2016) use Convolutional Neural Networks to a similar end. We build upon Lample et al.'s approach, replacing their CRF with a decoder RNN trained with an adjusted Actor-Critic objective.

We also apply the AC-RNN to CCG supertagging (Clark & Curran, 2004), where the model labels each word in a sentence with one of 1,284 morphosyntactic categories from CCGbank (Hockenmaier & Steedman, 2007). Bidirectional RNNs have been used in a number of recent supertagging systems (Lewis et al., 2016; Xu, 2016; Vaswani et al., 2016; Kadari et al., 2017; Wu et al., 2017). Among these, our effort is most similar to Vaswani et al. (2016), who also use an RNN decoder over a bidirectional encoder; however, they address exposure bias with Scheduled Sampling.

We also consider transduction tasks, which go beyond sequence labeling by allowing many-to-many monotonic alignments between the source and target symbols. In particular, we focus on transliteration, where the goal is to convert a word from a source script to a target script on the basis of the word's pronunciation. Many neural transliteration approaches follow the sequence-to-sequence model originally proposed for NMT (Jadidinejad, 2016; Rosca & Breuel, 2016). Our AC-RNN transliteration system is similar to these, but with an improved objective to address exposure bias.

Other approaches have been proposed to address exposure bias in RNN, especially for NMT. We review the following major techniques: Scheduled Sampling, Reinforcement Learning, and Imitation Learning. To our knowledge, none of these prior works has compared its method against CRF.

**Scheduled Sampling (SS):** Bengio et al. (2015) introduce the notion of scheduled sampling as the decoder RNN is gradually exposed to its own errors, where a sampling probability is annealed at every training epoch so that we use gold-standard inputs at the beginning of the training, but while approaching the end, we instead condition the predictions on the model-generated inputs.

**Reinforcement Learning (RL):** Ranzato et al. (2016) apply the REINFORCE algorithm (Williams, 1992) to Neural Machine Translation (NMT), to train the network with a reward derived from the BLEU score of each generated sequence. Bahdanau et al. (2017) apply the actor-critic algorithm in NMT by applying a reward-reshaping approach to construct intermediate BLEU feedback at each step. Rennie et al. (2017) introduce a Self-Critical (SC) training approach that does not require a critic model, which has been shown to outperform REINFORCE in the image captioning task. This method has also been applied to abstract summarization (Paulus et al., 2018).

Unlike these previous works that apply reinforcement-learning techniques to optimize an available external metric such as ROUGE in text summarization, or BLEU in translation, giving one reward at the end of each sequence, we demonstrate that sequence labeling tasks benefit from the binary rewards that are available at each step. In addition, Bahdanau et al. (2017), and Paulus et al. (2018) combine the Teacher Forcing maximum-likelihood objective with their proposed RL objectives, which requires two forward computations in the decoder RNN, one for conditioning on the ground-truth labels, another for the RL objective without conditioning on the ground-truth labels. In this work, we will incorporate the supervision of the gold label into the actor-critic algorithm itself without any extra computation. In contrast to Bahdanau et al. (2017), our method employs a simpler critic architecture, without any schedules to pre-train the critic model.

**Imitation Learning:**

Another alternative method to address exposure bias would be imitation learning, where one has access to a gold-standard policy instead of rewards. In the case of sequence labeling, this policy

corresponds to the gold-standard tag sequence, and most imitation-learning algorithms such as Dagger (Ross et al., 2011) reduce to variants of Scheduled Sampling. Recently, the related technique of learning to search (Daumé III et al., 2009) has been extended to neural network models by SEARNN (Leblond et al., 2018), but that has been shown to perform slightly worse than the actor-critic training for NMT.

## 3 ARCHITECTURE

### 3.1 ENCODER

Following Huang et al. (2015), the encoder of our sequence labeler employs a bi-directional RNN over the tokens in the sequence. The bi-directional RNN transforms context-independent token representations into representations of tokens-in-context, allowing each position to potentially encode information from the entire input sequence.

For sequence labeling, we follow Lample et al. (2016), and build our context-independent word representation by combining an embedding table with the outputs of a bi-directional RNN applied to each word's characters. The final states of the forward and backward character RNNs are concatenated to the word's embedding, and then passed through a dropout layer. We also concatenate capitalization pattern indicators to these feature vectors. Our word embeddings are initialized using embeddings pre-trained on a large corpus.

For transliteration, where we operate exclusively on the character level, we apply a bi-directional RNN on the character representations, which are provided by a randomly initialized embedding table.

### 3.2 DECODERS

Given an input $X = (x_1, x_2, ..., x_l)$, we look for an output sequence $Y = (y_1, y_2, ..., y_l)$ where each $y_t$ is an output token. In the encoder (Section 3.1), we transform the input $X$ into a sequence of hidden vectors $H = (h_1, h_2, ..., h_l)$. Given these vectors, the simplest decoder does not account for output dependencies at all. Instead, it independently predicts the output at time $t$ by mapping $h_t$ into a probability distribution $p_{\text{INDP}}(y_t|h_t)$ using a softmax layer. This results in a sequence-level probability of $p(Y|X) = \prod_t p_{\text{INDP}}(y_t|h_t)$.

In sequence labeling tasks, there are typically dependencies between the output tokens; for example, in English Part of Speech tagging, a determiner is unlikely to follow another determiner. The most prominent and widely used approach to track such dependencies is the CRF, which uses dynamic programming over an undirected graphical model to maintain a well-defined probability distribution over sequences, effectively modeling $p(Y|X) = p_{\text{CRF}}(Y|H)$, where $p_{\text{CRF}}$ hides fixed-order Markov dependencies over $Y$. The CRF can be used as a node in a neural sequence labeler, while still allowing the system to train end-to-end (Huang et al., 2015).

An alternative technique is to use a decoder RNN on top of the encoder, as is typically done in neural machine translation (Sutskever et al., 2014), and has been done for CCG super-tagging (Vaswani et al., 2016). Let $d_t$ be the recurrent decoder state, summarizing the output sequence up to time $t$, and let $c_t$ be the context vector that summarizes the input $X$ for time $t$. For sequence labeling, the source-to-target alignment is trivial, and $c_t$ is provided directly by the encoder: $c_t = h_t$. We can then use the input-feeding method of Luong et al. (2015) to define $d_t$ recursively: $d_t = \text{RNN}(d_{t-1}, y_{t-1}, c_{t-1})$, where RNN is a recurrent unit, in our case, an LSTM (Hochreiter & Schmidhuber, 1997). Finally, a softmax layer is used to define a probability distribution over output tokens $p_{\text{SM}}(y_t|c_t, d_t)$, resulting in a sequence model of: $p(Y|X) = \prod_t p_{\text{RNN}}(y_t|h_t, y_{t'<t}) = \prod_t p_{\text{SM}}(y_t|c_t, d_t)$. During training, the gold-standard previous token $y_{t-1}$ is fed into the decoder at time $t$, while at test time, we use the model's generated output.

For transliteration, where the input and output sequence lengths do not match, we can no longer simply provide the encoder state $h_t$ at time $t$ as the context vector $c_t$ for our probability models. Following standard practice in NMT, for the decoder RNN, an attention mechanism (Bahdanau et al., 2015) can learn an alignment model that provides a scalar score $\alpha_{t,t'}$ for each target position $t$ and source position $t'$, giving us a context vector $c_t = \sum_{t'} \alpha_{t,t'} h_{t'}$, which we can use in place of $h_t$

---

**Algorithm 1** adjusted Actor-Critic Training

---

- · Input: Source $X$, Target $Y$, and $n$ as hyper-parameter
- · Greedy decode $X$ using $\theta$ to get:
    - · the output sequence $\hat{Y} = (\hat{y}_1, ..., \hat{y}_l)$
    - · decoder RNN states $D = (d_1, ..., d_l)$
    - · context vectors $C = (c_1, ..., c_l)$
- · For each output target position $t$:
    - · $r_t = 1$ if $\hat{y}_t = y_t$, 0 otherwise
    - · $V_{\theta'}(t) = \text{CriticNetwork}(d_t, c_t, \theta')$
- · $loss_\theta = 0$; $loss_{\theta'} = 0$
- · For each output target position $t$:
    - · $G_t = \sum_{i=0}^{n-1} [r_{t+i}] + V_{\theta'}(t+n)$
    - · $\delta_t = G_t - V_{\theta'}(t)$
    - · $a\delta_t = \text{adjust}(y_t, \hat{y}_t, \delta_t) \times \delta_t$
    - · $loss_\theta = loss_\theta - a\delta_t \ln p_\theta(\hat{y}_t | X, \hat{y}_{t'<t})$
    - · $loss_{\theta'} = loss_{\theta'} + \delta_t \times \delta_t$
- · Back-propagate through $loss_\theta$ as normal to update $\theta$
- · Perform a semi-gradient step along loss $\theta'$ to update $\theta'$

---

in the RNN models described above. Specifically, we use the global-general attention mechanism of Luong et al. (2015). To modify the CRF for transliteration, we must allow it to generate output of a different length from its input. To do so, we pad both sequences with extra end symbols up to a fixed maximum length, and let CRF decode until the end of the padded source sequence. It controls its target length by outputting padding tokens.

All of the models described above can be trained with a maximum likelihood objective:

$$J_{ml}(\theta) = \sum_{X,Y} \ln p_\theta(Y|X)$$

## 4 ACTOR-CRITIC TRAINING

We adopt the actor-critic algorithm (Sutton & Barto, 1998; Konda & Tsitsiklis, 2003; Mnih et al., 2016) to fine-tune the decoder RNN. In AC training, the decoder RNN first generates a greedy output sequence according to its current model, similar to how it would during testing. We calculate a sequence-level credit (return) for each prediction by comparing it to the gold-standard. The AC update modifies our RNN to improve credits at each step. This process exposes the decoder to its own errors, alleviating exposure bias. Algorithm 1 provides pseudo code for the training process, which we expand upon in the following paragraphs.

We define the token-level reward $r_t$ as $+1$ if the generated token $\hat{y}_t$ is the same as the gold token $y_t$, and as 0 otherwise. We compute the sequence-level credit $G_t$ for each decoding step using the multi-step Temporal Difference return (Sutton & Barto, 1998):

$$G_t = \sum_{i=0}^{n-1} [r_{t+i}] + V_{\theta'}(t+n)$$

The step count $n$ allows us to control our bias-variance trade-off, with a large $n$ resulting in less bias but higher variance. The critic $V_{\theta'}(t)$ is a regression model that estimates the expected return $E[G_t]$, taking the context vector $c_t$ and the decoder's hidden state $d_t$ as input. It is trained jointly alongside our decoder RNN, using a distinct optimizer (without back-propagating errors through $c_t$ and $d_t$). With this critic in place, the update for the AC algorithm is defined as

$$\frac{\partial J_{ac}(\theta)}{\partial \theta} = \sum_t \frac{\partial \log(p_\theta(\hat{y}_t|X, \hat{y}_{t'<t}))}{\partial \theta} (\delta_t)$$

where:

$$\delta_t = G_t - V_{\theta'}(t)$$

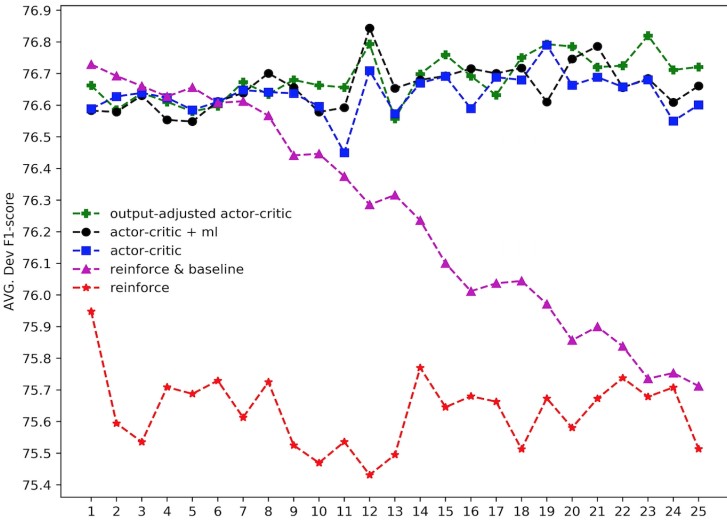

Figure 1: The adjusted actor-critic objective compared to alternative policy-gradient objectives on German NER with 17 possible output tags. All objectives are maximized using Gradient Ascent with a fixed step size of 0.5 for actor-critic, and 0.01 for REINFORCE objectives. For REINFORCE objectives, we set $n = l$ in the Temporal Difference credits (i.e. sum all rewards until the end of sequence). All methods are trained 20 times with different random seeds using the same hyper-parameters.

The AC update changes the prediction likelihood proportionally to the advantage $\delta_t$ of the token $\hat{y}_t$. Therefore, if $G_t > V_{\theta'}(t)$, the decoder should increase the likelihood. The AC error $\frac{\partial J_{ac}(\theta)}{\partial \theta}$ back-propagates only through the actor's prediction likelihood $p_\theta$.

**Critic Architecture**: We employ a non-linear feed-forward neural network as our critic, which uses leaky-ReLU activation functions (Nair & Hinton, 2010) in the first two hidden layers. In the output layer, it uses a linear transformation to generate a scalar value. To learn the critic's parameters $\theta'$, we use a semi-gradient update (Sutton & Barto, 1998). We do not use the full gradient in the Mean Squared error to train this regression model. Accordingly, for $\frac{\partial loss_{\theta'}}{\partial \theta'} = \frac{\partial(\delta_t \times \delta_t)}{\partial \theta'}$, instead of using $2\delta_t \frac{\partial(G_t - V_{\theta'}(t))}{\partial \theta'}$, we use the update $2\delta_t \frac{\partial V_{\theta'}(t)}{\partial \theta'}$. The Temporal Difference return $G_t$ uses the critic's estimate $V_{\theta'}(t + n)$. The full gradient will cause a feedback loop as by doing so, we will allow $G_t$ to match with $V_{\theta'}(t)$ in order to reduce the Mean Squared error.

**Adjusted Training**: Due to the inevitable regression error of the critic, and the fact that it is randomly initialized at the beginning, the advantage $\delta_t$ can undesirably become negative for a correctly-selected tag, or positive for a wrongly-selected tag. Optimizing the network according to these invalid advantages would increase the probability of the wrong tags, while decreasing the probability of the true tag. In such cases, to help critic update itself and form better estimates in the next iteration, we clip $\delta_t$ to zero by defining the adjusted advantage $a\delta_t$ as $\mathrm{adjust}(y_t, \hat{y}_t, \delta_t) \times \delta_t$ where:

$$\mathrm{adjust}(y_t, \hat{y}_t, \delta_t) = \begin{cases} 0 & \text{if} & \hat{y}_t = y_t & \& & \delta_t < 0 \\ 0 & \text{if} & \hat{y}_t \neq y_t & \& & \delta_t > 0 \\ 1 & otherwise \end{cases}$$

By setting the advantage $a\delta_t$ to 0, the $\mathrm{adjust}$ term effectively switches off the entire actor update when the advantage has the wrong polarity. Note that the critic is always updated.

Similar to prior RL works (Ranzato et al., 2016; Bahdanau et al., 2017; Paulus et al., 2018), we pre-train the network using the Teacher Forcing maximum-likelihood objective ($J_{ml}$). We then continue training from the best model using our adjusted actor-critic objective. Figure 1 shows development set performance, starting from the same $J_{ml}$-pre-trained point, for our adjusted objective, as compared to standard actor-critic, REINFORCE with baseline, and normal REINFORCE on German NER. We observe that the adjusted training helps the model reach a higher point compared to all

other objectives. Unlike prior RL methods, the adjusted actor-critic objective does not require any schedule for pre-training the critic. It also avoids the necessity of combining the actor-critic objective with the Teacher Forcing maximum-likelihood training, which is shown as actor-critic + ml in Figure 1. After the pre-training phase, the related works (Ranzato et al., 2016; Bahdanau et al., 2017; Paulus et al., 2018) still combine the maximum-likelihood training with their proposed RL objectives.

## 5 EXPERIMENTS

We comprehensively compare AC-RNN, CRF, and RNN on three tasks: NER tagging, CCG supertagging, and machine Transliteration. We then compare the adjusted actor-critic objective with Scheduled Sampling on NER. Finally, we compare our training strategy to the Self-Critical method of Rennie et al. (2017).

### 5.1 SETUP

**Datasets**: To conduct the NER experiments, we use the English and German datasets of the CoNLL-2003 shared task (Tjong Kim Sang & De Meulder, 2003). Both datasets are annotated with 4 different entity types: 'Location', 'Organization', 'Person', and 'Miscellaneous' (e.g. events, nationalities, etc.). As we have multi-word named entities (e.g. 'University of XYZ'), we employ the 'BILOU' tagging scheme (Ratinov & Roth, 2009).

For CCG supertagging, we use the English CCGbank (Hockenmaier & Steedman, 2007), the standard sections {02-21}, {00}, and {23} as the train, development, and test sets, respectively. We consider all the 1284 supertags appeared in the train set.

We use pre-trained, 100-dimensional *Glove* embeddings (Pennington et al., 2014) for all English word-level tasks, and fine-tune them during training. For German NER, we obtain the embeddings (64 dimensions) of Lample et al. (2016), which are trained on a German monolingual dataset from the 2010 Machine Translation Workshop. We apply no preprocessing on the datasets except replacing the numbers and unknown words with the 'NUM' and 'UNK' symbols.

We conduct the transliteration experiments on the English-to-Chinese (EnCh), English-to-Japanese (EnJa), English-to-Persian (EnPe), and English-to-Thai (EnTh) datasets of the NEWS-2018 shared task.[1] The training sets contain approximately 40K, 30K, 10K and 30K instances for EnCh, EnJa, EnPe, and EnTh, respectively, while the development sets have 1K instances. We train the models on the training sets, and evaluate them on the development sets. We hold out 10% of the training sets as our internal tuning sets.

**Training Details**: For the experiments, our different models share the same encoder, using the same number of hidden units (see the appendix for the hyper-parameters used in our experiments). The maximum-likelihood training is done with the Adam optimizer (Kingma & Ba, 2015) with a learning rate of 0.0005. The RL training is done with the mini-batch gradient ascent ($\theta = \theta + \alpha \frac{\partial J_{ac}(\theta)}{\partial \theta}$) using a fixed step size of 0.5 for NER & CCG, and 0.1 for Transliteration [2]. The critic is trained with a separate Adam optimizer with the learning rate of 0.0005. We employ a linear-chain first-order undirected graph in the CRF model.

As performance varies depending on the random initialization, we train each model 20 times for NER and 5 times for CCG using different random seeds which are the same for all models. We report scores averaged across these runs $\pm$ the standard deviations. Due to time constraints, for the transliteration experiments, we train each model only once.

**Evaluation**: We compute the standard evaluation metric for each task: entity-level F1-score for NER, tagging accuracy for CCG, and word-level accuracy for transliteration. For the models with decoder RNNs, we report the results achieved using a beam search with a beam of size 10. For the NER and CCG experiments, we conduct the significance tests on the unseen final test sets, using the Student's t-test over random replications at the significance level of $0.05$.

---

[1] http://workshop.colips.org/news2018/shared.html
[2] We also tried Adam and RMSProp optimizers for the RL training, but both completely diverged.

| Model | Dev (En) | Test (En) | Dev (De) | Test (De) |
|---|---|---|---|---|
| INDP | 93.63 ±0.13 | 89.77 ±0.21 | 75.51 ±0.28 | 72.15 ±0.57 |
| RNN | 94.43 ±0.16 | 90.75 ±0.23 | 76.85 ±0.39 | 73.52 ±0.36 |
| CRF | 94.47 ±0.12 | 90.80 ±0.19 | 76.27 ±0.35 | 73.59 ±0.36 |
| AC-RNN | **94.54** ±0.12 | **90.96** ±0.15 | **77.10** ±0.29 | **73.82** ±0.29 |
| Lample et al. (2016) | | 90.94 | | 78.76 |

Table 1: Average entity-level F1-score for English & German NER on the CoNLL-2003 datasets. Reimers & Gurevych (2017) report 90.81 as the median performance for the CRF model of Lample et al. (2016) in English.

| Model | Dev | Test | Memory (GB) | Time (m) |
|---|---|---|---|---|
| INDP | 94.24 ±0.03 | 94.25 ±0.11 | **1.3** | **5** |
| RNN | 94.25 ±0.06 | 94.28 ±0.09 | 1.8 | 11 |
| CRF | 94.31 ±0.07 | 94.15 ±0.11 | 9.8 | 50 |
| AC-RNN | **94.43** ±0.08 | **94.39** ±0.06 | 1.5 | 10 |
| Vaswani et al. (2016) | 94.24 | 94.50 | | |
| Kadari et al. (2017) | 94.37$^\diamond$ | 94.49$^\diamond$ | | |

Table 2: Average top-1 accuracy, and the required GPU memory and execution time (one epoch) on English CCG supertagging. The $^\diamond$ results are achieved with CRF training.

## 5.2 RESULTS

**Main Comparisons**: As our primary empirical study, we compare the AC-RNN to CRF and RNN models. we also consider independent prediction of the labels as another baseline.

The results of the NER experiments are shown in Tables 1. As expected, we observe that by modelling the output dependencies using either an RNN or a CRF, we achieve a significant improvement over the baseline INDP, about $1\%$ F1-score on both English and German datasets. With respect to prior work, our CRF model replicates the reported results on English NER. On German NER, we cannot replicate the CRF results of Lample et al. (2016), although we obtained their German word embeddings. We attribute this discrepancy to different preprocessing of the dataset. Moreover, AC-RNN significantly outperforms both RNN and CRF on both English and German test sets with the corresponding P values of 0.001 and 0.004 for RNN, and 0.003 and 0.016 for CRF. These results demonstrate that AC-RNN is successful at overcoming the RNN's exposure bias, and represents a strong alternative to CRF for named entity recognition.

On CCG supertagging (Table 2), AC-RNN is significantly better than all other models with the P values of 0.019, 0.025, and 0.002, respectively, and is competitive with reported state-of-the-art results. For this task, we had expected the improvements to be larger, because of CCG supertagging's potential for long-distance output dependencies. Instead, the results show that independent predictions do surprisingly well.

Table 2 also shows the time and memory requirements for each method on the CCG task. We observe that, due to the large output vocabulary size of the task (1284 supertags), CRF is five times slower than AC-RNN during training, while the batched version of its Forward algorithm requires six times more GPU memory during training. The Forward algorithm of CRF runs out of memory with the mini-batch size of 16 on a 12-GB Graphical Processing Unit.

The transliteration results in Table 3 show that AC-RNN outperforms CRF (likely due to CRF's inability to predict an output of a different length from its input), as well as RNN (likely due to its exposure bias). The transliteration experiments support our hypothesis that AC-RNN is more generally-applicable than CRF, and the improvements from the adjusted actor-critic training transfer to other tasks.

To confirm that our RNN baseline performs reasonably well, we validate our transliteration model against a standard NMT implementation as provided by the OpenNMT tool (Klein et al., 2017). We apply the tool "as-is" with its default translation hyper-parameters. Note that our RNN system in this experiment is also an NMT-style model with an attention mechanism.

| Model | EnCh | EnJa | EnPe | EnTh |
|---|---|---|---|---|
| CRF | 67.6 | 45.8 | 75.6 | 32.2 |
| RNN | 70.6 | 51.6 | 76.3 | 39.7 |
| SC-RNN | 70.2 | 51.8 | 77.2 | 41.3 |
| AC-RNN | **72.3** | **52.4** | **77.8** | **41.4** |
| OpenNMT | 70.1 | 47.7 | 70.5 | 36.3 |

Table 3: The word-level transliteration accuracy on the development sets of NEWS-2018 shared task. SC: Self-Critical training

| Model | Dev | Test |
|---|---|---|
| RNN | 76.85 $\pm$0.39 | 73.52 $\pm$0.36 |
| SS-RNN | 76.93 $\pm$0.32 | 73.65 $\pm$0.29 |
| SC-RNN | 76.71 $\pm$0.27 | 73.50 $\pm$0.42 |
| AC-RNN | **77.10** $\pm$0.29 | **73.82** $\pm$0.29 |

Table 4: The adjusted actor-critic training compared to Scheduled Sampling (SS-RNN), and Self-Critical training (SC-RNN) on German NER.

**Scheduled Sampling Comparisons**: In the next experiment, we compare the adjusted actor-critic objective to Scheduled Sampling. We implement this approach using the inverse sigmoid schedule of Bengio et al. (2015) for annealing $\epsilon$, and denote it as SS-RNN. Table 4 shows the comparison of SS-RNN with AC-RNN on German NER. Both systems improve over RNN, but AC-RNN is significantly better than SS-RNN with the P value of 0.040. This result supports our hypothesis that the reinforcement-learning solutions should outperform Scheduled Sampling, as the adjusted actor-critic training considers the entire sequence, whereas Scheduled Sampling addresses only exposure to the immediately previous token. We also observe that Scheduled Sampling, unlike the adjusted actor-critic training, is highly sensitive to the choice of sampling schedule (see the appendix for a quantitative analysis).

**Self-Critical Comparisons**: In our final experiment, we compare the adjusted actor-critic training with the Self-Critical policy training of Rennie et al. (2017) which does not require a critic model. This method is intended to represent the state-of-the-art in reinforcement learning for sequence-to-sequence models with sequence-level rewards, to be contrasted against our AC-RNN and its position-level rewards. The transliteration results shown in Table 3 indicate that the Self-Critical training improves over RNN on EnJa and EnPe, and EnTh, however, it fails to beat AC-RNN across all the evaluation sets. On German NER, the Self-Critical training cannot improve over RNN as shown in Table 4. This observation is aligned with our initial hypothesis that the reinforcement-learning techniques applied to sequence labeling would benefit more from modelling the intermediate rewards, as is done with the Temporal Difference credits in the adjusted actor-critic training.

## 6 CONCLUSION

We have proposed an adjusted actor-critic algorithm to train encoder-decoder RNNs for sequence labeling tasks. Though related reinforcement-learning algorithms have previously been applied to sequence-to-sequence tasks, our proposed AC-RNN is specialized to sequence-labeling by taking advantage of the per-position rewards. To our knowledge, we have presented the first direct, controlled comparison between CRFs and any form of RNN. On NER and CCG supertagging, our system significantly outperforms both RNN and CRF, establishing the AC-RNN as an efficient alternative for sequence labeling. We have also demonstrated the advantages of the AC-RNN in terms of its flexibility, fast training, and small memory footprint. Finally, we showed that our proposal for handling exposure-bias outperforms the related alternatives of Scheduled Sampling and Self-Critical policy training.

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

| Hyper-parameter | En/De | CCG | TL |
|---|---|---|---|
| char_embedding_size | 32 | 32 | 128 |
| output_embedding_size | 32 | 128 | 128 |
| max_gradient_norm | 5.0 | 5.0 | 10.0 |
| encoder units | 256 | 512 | 256 |
| decoder units | 256 | 512 | 256 |
| batch size | 32 | 10 | 64 |
| $n$ | 2/4 | 8 | 6 |
| dropout | 0.5 | 0.5 | 0.5 |
| RNN gate | LSTM | LSTM | LSTM |

Table 5: The hyper-parameters used in the experiments.

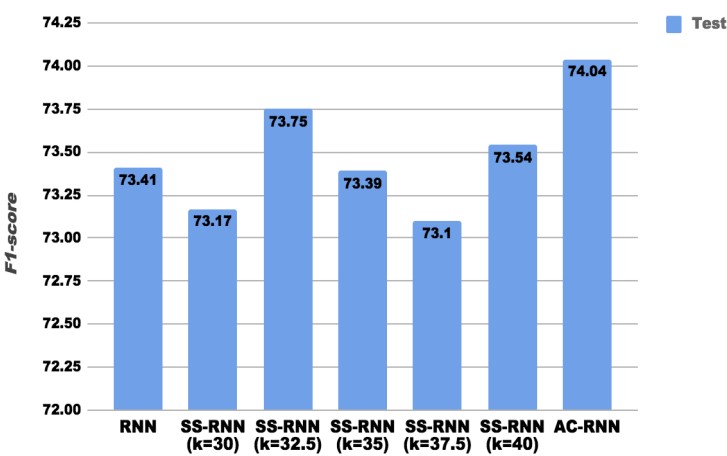

Figure 2: The sensitivity of Scheduled Sampling on the choice of sampling schedule on German NER. Higher $k$ results in less sampling.

## A   HYPER-PARAMETERS

Table 5 provides the hyper-parameters used in our experiments.

## B   SCHEDULE FOR SCHEDULED SAMPLING

Figure 2 illustrates that Scheduled Sampling is highly sensitive to the choice of sampling schedule.

