# OpenReview forum: "EFFICIENT SEQUENCE LABELING WITH ACTOR-CRITIC TRAINING"
_ICLR.cc/2019/Conference_

### Official Review · AnonReviewer1 · 2018-10-29
**Unclear focus of the paper: tagging or sequence generation? Comparisons not informative**

**Rating:** 4
**Confidence:** 5

**Review:**

I found the paper difficult to follow. The method proposed is not well motivated, and  the literature review explains well the novelty. Here are some questions/points for discussion:

- the token-level MLE training is not what causes the exposure bias: one can train with MLE and still avoid it by generating appropriate sequences using the RNN, as in scheduled sampling. The problem with MLE (or cross entropy) is that the labels to be predicted might not be the correct ones. See the paper by Ranzato et al. (ICLR2016) for a good discussion of the issue: https://arxiv.org/pdf/1511.06732.pdf

- The criticism against previous works for not comparing agains CRFs seems odd: CRFs are given the number of labels, words, etc. to predict, typically the same as the number of words to be tagged. If one  has this, as well as binary rewards for each decision, then there is little benefit for RL/IL based approaches to be used. The point for them is the use of non-decomposable loss functions such as BLEU, which are not common in tagging, but in tasks like MT, where CRFs can't be used. In fact, for the transliteration experiments in the paper, the CRF approach is padded to perform the task, which highlights that it is not the right comparison.

- the approach proposed seems very similar to MIXER, which also learns a regressor to predict the reward for each action. A direct comparison both in terms of how the approaches operate and empirically is needed.

- why is it a problem that previous works by Ranzato, Bahdanau and Paulus combine MLE and RL? You are using the same supervision, ie. the labeled corpus.

- the adjusted training seems to essentially not reward correct predictions (top branch in the equation). Why is this a good idea?

- In figure 1 it is not clear at all that the proposed approach works; depending on the epoch the ranking among the three variants differs


- what does it mean for one method to surpass the other in flexibilty? If anything the requirement for immediate rewards after every action restricts flexibility, as one can't use non-decomposable loss functions such as BLEU which are prety common in NLP.

- How is the training efficiency measured in the paper?

- Why not compare against MIXER, as well as more recent work by Leblonde et al. (2018): https://arxiv.org/abs/1706.04499 ? I don't see why the Rennie et al. 2017 method is picked for comparison.

- It is not true that in IL one needs a gold standard policy, one can learn with sub-optimal policies, see Sun et al. (2018): https://arxiv.org/pdf/1703.01030.pdf

- It is odd to say that an approach proposed earlier (Dagger) reduces to a variant of a later proposed one (Scheduled sampling), the reduction should be the other way around

- are the randomly initialized character embeddings for transliteration tuned during training?

- How were the alignments for training the CRF obtained?

---

> ### Author Response · Authors · 2018-11-13
> **Clarity**
>
> Thank so much for your insightful comments.
> -
> - Our main goal is to replace CRF in tagging, especially for tasks with large number of labels. `
>
> - MIXER uses REINFORCE, table 1illustrates that REINFORCE family fails compared to actor-critic.
>  We both have a regressor as critic, but MIXER doesn't bootstrap its estimates in the computed returns.
>
> - Combining MLE and RL requires two forward computation in decoder RNN, one conditioning on gold standard, another on model-generated tokens.
>
> - The prediction is correct, but the advantage given by the critic is wrong, so we skip the update on actor.
>
> - Average over 20 runs, on epoch 13, adjusted actor-critic are better than actor-critic and reinforce models.
>
> - Reward reshaping can be used to convert non-decomposable loss functions such as BLEU into step-size rewards.
> Reward(t) = BLEU(t) - BLEU(t-1)
>
> - End-to-End training time and the required GPU memory for one training epoch on CCG supertagging.
>
> - We tried the open source code for AC of Bahdanau on transliteration, but it completely failed.  We only obtained dev and test split from Leblonde et al. (2018) on spelling correction dataset.
>
> - We will review the mentioned paper.
>
> - Scheduled Sampling is inspired by Dagger.
>
> - Character embeddings are fine-tuned during training.
>
> - There is no preprocess alignment done for CRF in transliteration.

---

> > ### Comment · AnonReviewer1 · 2018-11-25
> > **Clarity**
> >
> > "- Our main goal is to replace CRF in tagging, especially for tasks with large number of labels. `"
> >
> > But then you should look at tagging tasks which is what (neural) CRF approaches are good for; transliteration, as pointed out in the introduction of the paper is not such a task.
> >
> > - "Combining MLE and RL requires two forward computation in decoder RNN, one conditioning on gold standard, another on model-generated tokens. "
> >
> > And why is this a problem? Assuming it is the extra computation, then there should be a careful comparison on processing time needed between the methods.
> >
> > "- Reward reshaping can be used to convert non-decomposable loss functions such as BLEU into step-size rewards. Reward(t) = BLEU(t) - BLEU(t-1)"
> >
> > But this is an approximation, so whether it will work depends on many factors. In any case, this sounds more like an argument on BLEU's flexibility not of the proposed method.
> >
> > -  " There is no preprocess alignment done for CRF in transliteration. "
> >
> > The paper states: "To do so, we pad both sequences with extra end symbols up to a
> > fixed maximum length, and let CRF decode until the end of the padded source sequence."
> > As CRF is a tagging method, somehow there is an alignment of input to output characters, which is why you need the padding.

---

### Official Review · AnonReviewer3 · 2018-10-31
**The paper pretenses reinforcement learning algorithms for dealing with the "exposure bias" problem of RNNs in sequence labeling tasks. The paper suffers from clarity issues - for example it was hard to understand the exposure bias problem. I also miss an important comparison in the experimental section - to the LSTM-CRF model.**

**Rating:** 4
**Confidence:** 3

**Review:**

The paper pretenses reinforcement learning algorithms for dealing with the "exposure bias" problem of RNNs in sequence labeling problems.  While I admire the thoroughness of  both the algorithmic work and experimental setup, I am afraid the paper suffers from two major problems:

1. The paper suffers from serious clarity issues. Particularly, the main problem the paper deals with - exposure bias- is not well explained. I admit that while I am working with RNNs on a regular basis, I was not familiar with this problem. Unfortunately, I was also not able to understand it from the paper.  This may be a very basic concept, but a paper must be self-contained. Unfortunately, after reading the paper, front to cover, I cannot tell what is the problem the authors are trying to solve (except, of course, from providing a better training algorithm for RNNs).

2. As the authors say already in the abstract, one of the best performing models on structured NLP tasks is LSTM-CRF, which combines the power of both the neural and the structured prediction frameworks. However, the authors do not compare their solution to LSTM-CRF, but only to LSTM and to CRF. This is a very important baseline, and without a proper comparison it is hard to evaluation the contribution of this paper.

---

> ### Author Response · Authors · 2018-11-06
> **Clarity**
>
> 1- The concern is well understood, though related works have already defined it clearly. The bias is originated from the method of training, not the RNN itself.
>
> 2-Throughout the paper:
>
> CRF: LSTM encoder + CRF decoding with MLE training
> RNN: LSTM encoder + LSTM decoder with MLE training
> AC-RNN: LSTM encoder + LSTM decoder with MLE & Actor-Critic training

---

### Official Review · AnonReviewer2 · 2018-11-02
**Actor critic for sequence labeling; not very novel but good results on transliteration; inadequate comparison**

**Rating:** 5
**Confidence:** 4

**Review:**

The authors propose actor-critic method for sequence labeling and show that it performs better (is more stable than) other RL approaches and also outperforms other techniques for countering exposure bias like scheduled sampling.

The results show very small improvement in tagging tasks like NER and CCG supertagging compared to other approaches ; but they show good improvement in the transliteration task which is more of a transduction task than a tagging task.

This authors also discus the adjusted training procedure which accounts for bad performance of the critic model in the initial stages of training. The approach is not very novel because actor-critic for more general sequence-to-sequence models (arguably more complex than tagging) has already been explored in the literature (Bahdanau et al., cited by the authors). Major difference in the proposed approach is the use of stepwise hamming-loss based reward and it is unclear whether this is a major contribution which  sets it apart from the previous work on AC for sequence modeling. For example, a good comparison would be to do tagging in seq2seq style and use the approach proposed in the existing AC work to show the value of the approach proposed here.

Also, minor claims about thoroughness of comparison with CRF are ill-founded as previous published work on tagging has indeed compared CRFs, independent, LSTM/RNN based models.

---

> ### Author Response · Authors · 2018-11-06
> **Clarity**
>
> Our main goal is to replace CRF in tagging, especially for tasks with large number of labels. `
> We would appreciate if we can be referred to an existing paper comparing Seq2Seq with encoder RNN + CRF decoding.

---

### Meta-Review · Area_Chair1 · 2018-12-14
**mismatched goals, evaluation and comparison**

**Confidence:** 5
**Recommendation:** Reject

**Metareview:**

this is an interesting approach to use reinforcement learning to replace CRF for sequence tagging, which would potentially be beneficial when the tag set is gigantic. unfortunately the conducted experiments do not really show this, which makes it difficult to see whether the proposed approach is indeed a viable alternative to CRF for sequence tagging with a large tag set. this sentiment was shared by all the reviewers, and R1 especially pointed out major and minor issues with the submission and was not convinced by the authors' response.